# The Effects of CuO and SiO₂ on Aluminum AA6061 Hybrid Nanocomposite as Reinforcements: A Concise Review

**Muntadher Sabah Msebawi** [1,*] , **Zulkiflle Leman** [1,2,*], **Shazarel Shamsudin** [3], **Suraya Mohd Tahir** [1], **Che Nor Aiza Jaafar** [1], **Azmah Hanim Mohamed Ariff** [1], **Nur Ismarrubie Zahari** [1] **and Mohammed H. Rady** [4]

1   Department of Mechanical and Manufacturing Engineering, Faculty of Engineering, Universiti Putra Malaysia, Serdang 43400, Selangor, Malaysia; su_mtahir@upm.edu.my (S.M.T.); cnaiza@upm.edu.my (C.N.A.J.); azmah@upm.edu.my (A.H.M.A.); rubie@upm.edu.my (N.I.Z.)
2   Advanced Engineering Materials and Composites Research Centre, Faculty of Engineering, Universiti Putra Malaysia, Serdang 43400, Selangor, Malaysia
3   Sustainable Manufacturing and Recycling Technology, Advanced Manufacturing and Materials Centre (SMART-AMMC), Universiti Tun Hussein Onn Malaysia, Batu Pahat 86400, Johor, Malaysia; shazarel@uthm.edu.my
4   Engineering Technical College (ETCN), Al-Furat Al-Awsat Technical University (ATU), Kut 52001, Iraq; mradhi@uowasit.edu.iq
*   Correspondence: muntadharsabah86@gmail.com (M.S.M.); zleman@upm.edu.my (Z.L.)

**Abstract:** Hybrid composites are obtained by embedding multiple micro and nano reinforcements into the matrix materials. These hybrid composites are helpful to obtain the useful properties of matrix and reinforcement materials. Aluminum matrix is one the most common matrix materials due to its excellent thermal and electrical properties. This review covers various aspects of nanoparticle-reinforced Al hybrid composites. Solid-state recycling of Al only consumes around 5% of the energy utilized in the conventional extraction and recycling methods. This review revolves around the induction of silica and copper oxide nanoparticles into the solid-state recycled Al matrix material to form the hybrid composite. These nanoparticles enhance stiffness, toughness, and high temperature stability for Al hybrid composites. A detailed analysis was carried out for AA6061-grade Al matrix materials along with the silica and copper oxide nanoparticles. The present work focused on the effects of nano silica and nano copper oxide particle reinforcements on Al-based composite manufactured via hot extrusion process. The composite fabrication through solid-state recycling is discussed in detail. A detailed analysis for the effects of volume fraction and wt.% of CuO and SiO₂ reinforcement particles was carried out by various characterization techniques. A detailed comparison in terms of mechanical performance of Al-based composites with the addition of nano silica and nano copper oxide particles is presented here to investigate the efficiency and performance of these particles.

**Keywords:** aluminum alloy 6061; bibliometric analysis; extrusion; hybrid composite

## 1. Introduction

In this modern era, a great number of resources are being used to develop materials with enhanced mechanical, physical, thermal, chemical, and electrical properties [1]. Hybrid composite materials are the latest generation of polyphase heterogeneous materials that consist of at least two or more reinforcements [2–5].

The final properties of the composites are determined by the constituent phase materials and interfacial properties reinforcement/matrix material compatibility, relative quantities, and fabrication processes [6–8].

Ceramic reinforcements enhance the high-temperature stability of the composite as most of the metallic matrix materials are stable up to moderate temperature. High temperatures bring structural changes in metallic structures and adversely affect the

mechanical properties [9–11]. Hybrid composites are essentially used to achieve high strength and weight reduction [12]. Metallic hybrid composite materials are used in the manufacturing of automobile parts such as drive shafts, cylinders, pistons, and brake rotors [13,14]. The classification of composite materials is shown in Table 1.

**Table 1.** Classification of matrix materials and reinforcements.

| Type | Matrix Material | Number of Reinforcements | Reinforcement Type | References |
|---|---|---|---|---|
| Composites | Polymer, metal, and ceramics | Single | Fiber, whiskers, particles, and layer reinforcements | [10,15] |
| Hybrid composites | Polymer, metal, and ceramics | Multiple | Fiber, whiskers, particles, and layer reinforcements | [2,4,11] |
| Nanocomposites | Polymer, metal, and ceramics | Single or combination of two or more reinforcements | Nanofibers, nanoparticles, and quantum dots | [16–18] |

Reinforcements in Al improve the overall physical and mechanical properties. An increase in yield and tensile strengths was observed and reported by Karakoç et al. and Karabulut et al. [19,20] for compacted reinforced aluminum. This increase was a result of limiting the mobility of dislocations and precipitation phases. In addition, the formation and size of precipitation phases in the composite increased the toughness with the inclusion of the reinforcements [13,21,22]. The addition of nanoparticles was reported to be used in producing different types of MMC products [23]. The choice may be informed by the particle size that promotes compatibility [24].

The review covers various aspects and mechanical properties of silica and copper oxide Al hybrid composite materials. A detailed analysis is carried out by analyzing various research studies on these nanoparticles. Due to the excellent mechanical and physical properties of CuO and $SiO_2$, these particles are commercially used in the fabrication of Al-based composites [22]. The addition of copper oxide and $SiO_2$ nanoparticles into the aluminum matrix has some limitations such as poor dispersion and formation of agglomerates [18]. An increase in silica concentration may lead to a decrease in the tensile properties [25]. Cu addition reduces the melting point and results in the formation of $Al_2Cu$ phase which enhances the strength of Al matrix [26].

The present work focuses on the development of nano silica and nano copper oxide particles reinforced AA6061-based composite. The composite fabrication through the hot extrusion is discussed in detail. The effect of volume fraction and wt.% of these reinforcement nanoparticles were analyzed through various characterization techniques. A detailed comparison in terms of mechanical performance of AA6061-based composites with the addition nano silica and nano copper oxide particle was carried out to investigate the efficiency these particles.

### 1.1. Aluminum and Alloys

Our planet has an ample amount of aluminum. Aluminum is alloyed with other materials to enhance the overall mechanical and physical properties. Aluminum-based alloys have lower densities, higher thermal and electrical conductivities, and higher resistance to corrosion and wear [27]. Due to high malleability, Al is easily formable with the use of various metallic working techniques. Moreover, Al can easily be casted, welded, and machined for a variety of different applications. This high malleability property makes Al cost-effective and economical in machining and forming applications [28]. Table 2 presents some of the most important properties of pure aluminum at room temperature [24,29].

**Table 2.** Different physical properties of pure aluminum at room temperature.

| | | | |
|---|---|---|---|
| **Melting Point** | 660 °C | **Yield Strength** | 35 MPa |
| **Boiling point** | 2470 °C | **Hardness** | 90 HB |
| **Density** | 2.71 g/cm$^3$ | **Elongation** | 40% in 50 mm |
| **Tensile strength** | 90 MPa | **Elastic modulus** | 69 GPa |
| **Poison's ratio** | 0.33 | **Shear modulus** | 25 GPa |

Aluminum and alloys are among the top non-ferrous metals used in numerous engineering applications [13]. The Aluminum Association (AA) is the coordinating body for aluminum and alloys. The association is in collaboration with aluminum product manufacturers assign different grades to Al and its alloys. Figure 1 shows different AA grades used in different industries. AA6061 is one of the most popular and common AA grades used in various applications ranging from aerospace, marine, to automobile industries. AA6061 consists of magnesium (Mg) and silicon (Si), which are precipitated in the Al matrix [30]. The addition of copper (Cu) or zinc (Zn) improves the strength of the alloy, while Si gives rise to brittle nature in the alloy. Mechanical properties of Al-based alloys are dependent on the manufacturing, heat treatments, and the composition of alloying elements such as Si, Cu, and Mg. The chemical constituent of aluminum alloy is shown in Tables 3 and 4 [12,31].

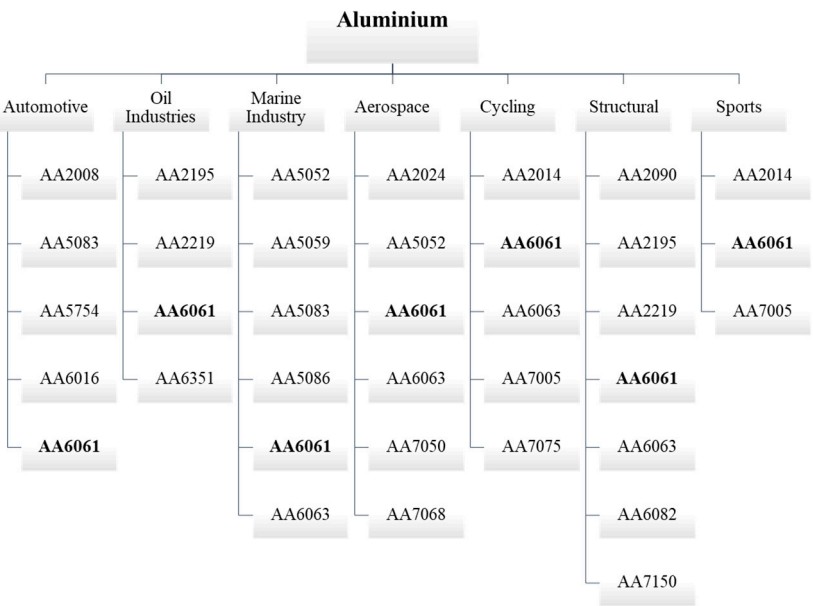

**Figure 1.** Various grading of AA used in different industries.

**Table 3.** Chemical composition of series 6000 aluminum alloy.

| Element | wt.% |
|---|---|
| Mg | 1.5–2 |
| Cu | Up to 2 |
| Mn | Up to 1.5 |
| B, Ti, Zr | Up to 0.3 |
| Si | 0.2–2 |
| Cr | Up to 0.5 |
| Zn | Up to 2.5 |
| Pb, Bi | Up to 1 |

**Table 4.** Chemical composition of series aluminum alloy.

| Al Alloy | Composition wt.% | | | | | |
|---|---|---|---|---|---|---|
| | **Si** | **Fe** | **Mn** | **Mg** | **Zn** | **Al** |
| AA6060 | 0.45 | 0.2 | - | 0.5 | - | Balance |
| AA6082 | 1.0 | 0.3 | 0.7 | 0.9 | - | Balance |
| AA3003 | 0.5 | 0.5 | 1.3 | - | - | Balance |
| AA5182 | 0.1 | 0.2 | 0.4 | 4.5 | - | Balance |
| AA5745 | 0.3 | 0.2 | 0.4 | 3.2 | - | Balance |
| AA6061 | 1.25 | 0.3 | - | 0.5 | - | Balance |
| AA7072 | 0.2 | 0.3 | 0.3 | 1.2 | 4.5 | Balance |

The choice of any Al-based alloy grade is dependent on the intended applications and required properties. Various Al alloy grades are currently being used in different applications and industries, as shown in Figure 1. The pros and cons of each Al alloy must be taken into account before using it in any application.

### 1.2. Overview of Aluminum Recycling Techniques

Aluminum is the second most found and used element on earth. Pure Al is the most used non-ferrous element in packaging, automobiles, construction, wires, and medicines. Aluminum oxide is extracted from bauxite ore through the Bayer process, and the Hall–Héroult process converts and refines aluminum oxide into Al through electrolysis. The advancement in the extraction technologies through research and development has resulted in the optimization of process parameters to extract pure Al but the basics of the process remained the same [32]. Recycling materials saves a huge amount of energy used in production. It is estimated that recycling aluminum scraps into ingots require around 5% of the energy that is utilized in extracting aluminum from bauxite to aluminum [33]. Recycling of materials is necessary to address the environmental concerns, to save energy and to decrease the pollution caused during the manufacturing processes of these materials [32]. Recycling of Al is a secondary production route, while the primary production of Al consists of a Bayer and Hall–Héroult process. About 35% of Al is recycled in comparison with 30% for zinc and 40% for copper and steel, while the energy saving of 95% for Al recycling is the highest among these materials. It is estimated that one ton recycled Al saves bauxite ores of around 8 metric tons, energy up to 14,000 kWh, around 6300 L of oil, and 7.6 m$^3$ of landfill space [34]. The main issue in the recycling of Al is to exclude various oxide particles and other impurities. The concentration of alloying elements can be controlled even after the recycling. The range of alloying elements in Al alloys can vary between 10 to 20 wt.%. The concentration of alloying elements can be altered during further processing of recycled Al. Various processes such as heat treatments, sorting, and comminution are employed to improve and enhance the quality of recycled Al. These processes increase the overall cost of the recycling process but with the significant increase in the quality of produced Al [35]. Figure 2 explains the processes involved in recycling of scrapped Al.

### 1.3. Solid-State Recycling

Solid-state recycling is a direct method to produce recycled Al from aluminum scrap. The recycling method reduces overall process cost with lower energy consumption and minimum environmental impact. Conventional recycling and melting processes are energy extensive and inefficient. These recycling methods undergo a material loss of up to 50% during the processing that increases overall costs and reduced efficiency [36]. The material loss is dependent on the factors such as heat treatment process, type of furnace used for melting, and scrap type [36]. Moreover, a considerable amount of material is lost due to oxidation during the melting process. These days, more interest is being shown towards the solid-state recycling, a solidus and melt less method, to save the material loss and to produce the final products with the exclusion of the melting process altogether [37].

During solid-state recycling, these materials undergo plastic deformation below the solidus temperature that removes the oxide layers from the surface. Due to the plastic deformation, fine grain size materials with enhanced mechanical properties are obtained. Hot extrusion, friction extrusion, rolling, forging, conforming, and ECAP are some of the methods used for solid-state recycling [37]. Parameters that must be taken into account before choosing any method include the size of chips, cold pressing parameters, extrusion temperature, extrusion rate, and ratio. These factors are the process indicators that must be controlled in order to increase the efficiency of the solid-state recycling process and mechanical properties of the recycled materials [38,39]. Several research studies of solid-state recycling techniques have found these methods to be far more economical and energy-efficient than conventional recycling techniques, but lack of control over properties in recycled materials makes it not feasible for mass-scale production [40]. Conventional recycling techniques have better control over mechanical, physical properties, microstructure, and chemical composition, while the efficiency of these methods is significantly lower than solid-state recycling [41].

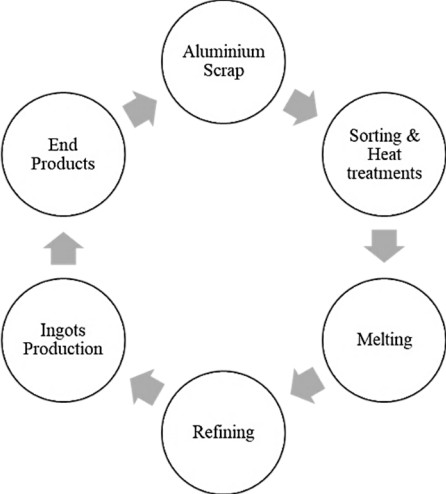

**Figure 2.** The processes in the recycling of Al scrap.

### 1.4. Metal Matrix Composite Materials

Three basic matrix materials are used in composites are metals, polymers, and ceramics. Metal matrix composites (MMCs) are formed by a combination of metallic matrix material and single or multiple reinforcements [41]. MMCs have matrix and reinforcement materials with different physical and mechanical properties combined to form a composite with a combination of useful properties from both materials [2,42]. MMCs have properties such as high stiffness and strength, higher wear resistance, lower coefficients of thermal expansions, and higher thermal conductivities [43]. Aluminum matrix composites (AMCs) attract great attention due to their lightweight and excellent mechanical properties. Due to the excellent strength to weight ratio, AMCs are employed in many high-end applications such as automobiles and aerospace engineering [44]. AMCs are used in the manufacturing of cylinder heads, crankshafts, pistons, engine blocks, and disk brakes. Due to their lightweight nature, AMCs are popular in the fabrications of various components of aircraft, spacecraft, transportation, and helicopters due to their excellent strength [43,45]. Required mechanical and physical properties are dependent on several factors such as individual properties of matrix and reinforcement materials, the orientation of reinforcements, matrix/reinforcement compatibility, and choice of fabrication method [46].

### 1.4.1. Matrix Materials

A matrix material is the most important constituent of the composite material. The selection of matrix material is dependent on the desired applications and properties as it

dictates the final properties of a composite. Matrix material incorporates the reinforcing materials to form the final composite structure and protects the reinforcements from any possible damage. The combination of both matrix and reinforcement materials assists in the uniform transfer of stress across the composite. Non-uniform and improper transfer of stress leads to the fracture and failure of the composite [23]. The stress transfer is strongly dependent on the matrix reinforcement interfacial properties and compatibility [47]. Al matrix materials with the combination of ceramic reinforcements provide enhanced and improved mechanical and physical properties for various applications [48].

1.4.2. Nano Reinforcement Materials

Wide attention is being given to incorporate nanomaterials into metallic composite matrix materials [49]. General advantages of nanoparticles include high surface area, high aspect ratio, and high surface energy. Nanoparticles are used as reinforcement due to exceptional structural and functional properties. As particle size is reduced, the interaction of the particle with impurities and dislocations becomes significant. Nanoparticle reinforcements prevent crack propagation and impart rigidity to the microstructures. Nanocomposites have at least one reinforcing material in the nanometric size range. Ravikumar et al. [47] found that ceramic particles offer higher mechanical properties than other monolithic metals. Table 5 shows various ceramic materials that have been widely used as reinforcements in metal matrix composites.

**Table 5.** Ceramic reinforcement materials.

| Ceramic Materials | |
|---|---|
| $SiO_2$ | $Al_2O_3$ |
| CuO | MgO |
| $B_2O_3$ | $Li_2O$ |
| $ZrB_2$ | $Fe_2O_3$ |
| $TiO_2$ | TiC |
| $K_2O$ | $B_4C$ |
| SiC | ZrC |
| $TiB_2$ | $ZrO_2$ |

The combination of two or more nano-, micro-, or macro-materials optimizes the physical and mechanical properties of the composite [4]. The addition of nanoparticles in the composite materials increases the overall fracture toughness, elastic modulus, damping, wear-resistance, and creep [12,50].

The inclusion of nanoparticles improves properties such as tensile and compressive behavior, ductility or elongation to failure, high-temperature mechanical properties, creep, dynamic mechanical properties, wear resistance, coefficient of thermal expansion, machining, fire resistance, and corrosion resistance. The improvement in various properties of MMCs due to nanoparticle inclusion is a result of factors such as selection of matrix and reinforcement, size of nanoparticles, compatibility of nanoparticles with the matrix material, selection of fabrication techniques, and heat treatment. Nano-MMCs have various advantages over micro-MMCs in many applications due to their superior combination of properties [22]. Some of the limitations associated with the use of nanoparticles as reinforcements are wettability issue with matrix materials, poor dispersion, and formation of agglomerates.

Effect of Silica ($SiO_2$) on Aluminum Alloy

It has been a challenge to fabricate Al-based alloys and composites with optimized dispersion, bonding, and distribution of ceramic reinforcements. Different research studies analyzed the effect of various ceramic particles such as $Al_2O_3$, $TiB_2$, SiC, and $ZrO_2$. However, only very few studies have focused on $SiO_2$ ceramic particles. The silica particles in Al matrix are added during casting in the alloys. The fluidity of the molten alloy is dependent

on factors such as surface energy, composition, viscosity, solidification rate, shape, size, and weight fraction of reinforcements [51].

In an experiment conducted by Zhu et al. [52], 30 wt.% micro-particles of $SiO_2$ were added to fabricate an in situ Al-Si/$\alpha$-$Al_2O_3$ composite at 677 °C via exothermic dispersive fabrication through powdered blends of silica and Al. Huo et al. [51] analyzed in situ fabrication of silica and Al-based composite. Silica and Al fine particles were obtained using ball milling. Issa et al. [25] fabricated Al–$SiO_2$ nano-composites with the use of hot extrusion and powder metallurgy. The samples had 1, 2, and 3 wt.% nano-silica. Even with the addition of 1 wt.% silica nanoparticle in the aluminum matrix, the mechanical properties of the composite were improved significantly, and the hardness and tensile strength of the nano-composite were enhanced by around 41.8% and 24.8%, respectively. Table 6 summarizes the fabrication methods and key finding of various studies on Al and silica composites.

**Table 6.** The key findings and fabrication methods of silica and Al based composites.

| MMC | Reinforcement | Fabrication Method | Key Findings | Reference |
|---|---|---|---|---|
| Al–$SiO_2$/$\alpha$-$Al_2O_3$ | 30 wt.% $SiO_2$ micro-particles | In situ powder dispersion method | Tensile strength and elongation increased with the addition of silica. The formation of $Al_2O_3$ limited the deformation process of the composite. Dislocation motion was restricted with the formation of different phases, which contributed to strengthening of the composite. | [52] |
| Al–$SiO_2$ | 10 wt.% $SiO_2$ micro-particles | In situ low temperature sintering | Fine particles of Al and $SiO_2$ were obtained via ball milling and diffusion couples were formed. $SiO_2$ particles were embedded in the matrix uniformly due to fine refining of particles through ball milling. The sintering process increased the relative density of the samples due to the homogenous distribution of $Al_2O_3$ particles. | [51] |
| Al–$SiO_2$ | Up to 3 wt.% addition of silica nanoparticles | Sintering and hot extrusion | Nanoparticles of silica and cold welding of Al particles were observed due to ball milling. Cold welding resulted in the homogeneous distribution of reinforcing nanoparticles. Differential thermal analysis revealed the stability of silica nanoparticles up to 900 °C. The porosity of the composite increased due to silica while the agglomerates and higher surface area of silica prevented the integration of Al particles. Hardness values of the composite increased significantly due to the homogenous distribution of ceramic silica nanoparticles. | [25] |
| Al–$SiO_2$ | Up to 19 wt.% addition of micro silica particles | Double stir casting method | The density of the composite decreased with the increase in wt.% of silica particles. Fine and refined particles of Al and silica increased the overall strength of the composite. EDX analysis showed the uniform distribution of reinforcing particles throughout the composite. The porosity of the samples was believed to be induced gases and moisture content during stir casting and degassing the samples would reduce the porosity. The hardness of the samples increased up to 22% with the increase in wt.% of silica mainly due to lesser defamation and good interfacial adhesion of the composite. Up to 11 wt.% of silica, ultimate tensile strength increased around 32%. | [53] |

Effect of Copper Oxide (CuO) on Aluminum Alloy

Copper has numerous applications and is one of the most widely used materials in the electronics industry due to the higher conductivity, oxide resistance, corrosion resistance, and thermal properties. However, copper has lower strength and poor wear resistance properties [54,55]. Metallic matrix material is reinforced with ceramic nanoparticles such

as borides, oxides, and carbides, and these nanoparticles enhance the high-temperature properties, stiffness, stability, and abrasion resistance of composites without compromising on the electric and thermal properties of Cu [56]. Cu–alumina composites have been extensively studied, and various research studies have reported the increase in hardness, tensile properties, strength, and erosion resistance with the addition of alumina [57–59].

Different research studies have focused on the Al–Cu-based composites with the addition of nanoparticles such as alumina, SiC, $SiO_2$, zirconia, magnesium oxide, and CNTs [60,61]. Raju et al. [62] fabricated and studied the properties of aluminum reinforced with powdered copper MMC. The composite was fabricated using the stir casting technique. The authors utilized 5 to 10 wt.% of copper for making composite samples. Cu powder was uniformly distributed across the composite samples. Plastic deformation and surface wear were observed for 5 wt.% copper sample, while 10 wt.% copper sample had better wear resistance, mainly due to the excellent refined matrix material and reinforced particles.. The final composite had uniform dispersion of Cu particles and had improved hardness, strength, and wear resistance [63,64] (Table 7).

**Table 7.** The key findings and fabrication methods of Cu- and Al-based composites.

| MMC | Reinforcement | Fabrication Method | Key Findings | Reference |
|---|---|---|---|---|
| Al–CuO | Up to 3 wt.% CuO nanoparticles | Stir mixing and squeeze casting | Ultimate tensile strength was enhanced with the increase in CuO nanoparticles. Agglomerates of CuO nanoparticles gave rise to stress concentration leading to the crack propagation and ultimately failure of the composite. At 3 wt.% concentration of CuO nanoparticles, ultimate tensile showed decreased due to the formation of the agglomerates at grain boundaries, giving rise to the brittle nature in the composite. | [64] |
| Al–Si/$CuAl_2$ | Up to 15 wt.% Cu microparticles | Squeeze casting | The hardness of the composite improved with the addition of Cu particles, mainly due to the presence of Si in Al matrix and refined grains. Hardness values slightly decreased after 6 wt.% Cu in the composite. Cu fine particles increased the strength and hardness of the composite. Homogenous dispersion of Cu particles enhanced the wear resistance. Heat treatment promoted the formation of $CuAl_2$ intermetallics. An increase in heat treatment temperature decreased the hardness values. | [63] |
| Al–$Mg_2$Si–Cu | Up to 4.5 wt.% Cu micro particles | Hot extrusion | Microstructure revealed the formation of intermetallic compounds in the alloy. Heat treatment enhanced the coefficient of friction for the alloy but was decreased with the increase in Cu concentration. Wear resistance and hardness increased with the increase in Cu concentration and heat treatment. | [65] |
| Al–Cu | Cu 20, 50, 80 vol.% | Spark plasma sintering | Ball milling incorporated deformations and strain in Al–Cu particles. Al underwent more plastic deformation due to ductile nature. Spark plasma sintering enabled the having of more control over the microstructure and mechanical properties of the composite. Composite samples had higher hardness values than respective materials due to the formation of intermetallics. | [66] |

Factors and Challenges Affecting Properties of MMCs

The parameters and issues that must be considered while fabricating new metallic composites are discussed in this section. Metallic composites have a various high-end application such as space and aircraft industries. Some of the factors that affect the physical and mechanical properties of the composites are discussed in Table 8. Factors influencing mechanical and physical properties of MMCs are nature of matrix and reinforcement, particle

size and aspect ratio of reinforcements, matrix/reinforcement compatibility, reinforcement concentration, dispersion and orientation of reinforcements, heat treatments, physical and chemical modification methods for matrix/reinforcements, and process parameters [67].

**Table 8.** Factors affecting physical and mechanical properties of metallic composites.

| Factors | Challenges | Effect on Properties | Reference |
|---|---|---|---|
| Wight and volume percentage | The volume percentage (vol.%) affects mechanical properties more significantly in comparison with wt.%. | Both wt.% and vol.% can be evaluated using various theories and mathematical calculations. Vol.% and wt.% of reinforcements affected hardening and strengthening mechanisms. | [6,68] |
| Reinforcement dispersion and orientation | Reinforcements can have improper distribution. Homogenous distribution ensures an increase in mechanical properties. Similarly, the orientation and compatibility of reinforcement fibers influence the properties of the composite. | Proper distribution of reinforcements was required for uniform load distribution in the composite. Agglomeration of reinforcements decreased mechanical properties due to improper load distribution. | [25,47] |
| Reinforcement size and shape | The shape and size of reinforcement affect the microstructure of MMCs | Mechanical properties of nano MMCs were significantly improved with the decrease in particle size. Shape, size, orientation, and coating over particles played a vital role in the determination of porosity and ductility. The finer particles increased the strength and hardness along with the cost reduction of the process. | [5,15,69,70] |
| Manufacturing and fabrication processes | Each processing method results in different microstructures and mechanical properties. | Extrusion force and pressure, die length, shrinking of composite, degassing, length and size of reinforcement, compatibility of matrix/reinforcement, the concentration of reinforcements, and flow patterns were important parameters to be considered in the manufacturing of the composite. | [71] |
| Porosity | Porosity affects the mechanical properties, reinforcement/matrix interfacial properties, and strengthening of the composite. Degassing of the samples can reduce the porosity. | Mechanical properties were improved with the decrease in porosity. Pores act as stress concentration sites and propagate cracks, which will lead to failure and fracture of the composite. Pores decrease the interfacial properties of the composite. | [25,72,73] |
| Interfacial bonds | Weak interfacial bonding and properties prevent achieving the desired mechanical properties. Good reinforcement/matrix compatibility gives rise to strong interfacial bonds. | A strong interfacial bond between the matrix and reinforcement can effectively and uniformly distribute the load in the composite. The pores act as sites for crack initiation, resulting in a decrease in overall tensile strength. Interfacial bonds can be improved through various physical and chemical modification techniques. | [5,74] |

## 2. Fabrication Methods

### 2.1. Fabrication of MMCs

The manufacturing process ensures the dispersion and distribution of reinforcements in the matrix material for better mechanical properties to be used in various applications [7]. Different melt less fabrication techniques are used for MMCs such as hot extrusion, friction extrusion, cold rolling, forging, and sintering [47]. The factors influencing the final properties of MMCs are categorized into material, design, and process parameters. Material parameters include the type of matrix and reinforcement material, morphology, size, and properties of these materials. Design parameters include ensuring the optimized design of the process to obtain the desired properties in final products. Process parameters include processing temperature, heat treatments, and material cleaning. Hot extrusion will be

discussed in detail for Al-based composite. Hot extrusion is one of the most effective methods to recycle Al-based composites [75–77].

### 2.2. Hot Extrusion Process

Extrusion is a plastic deformation process that involves the flow of solid billets through a desired cross-sectional area opening. Hot extrusion as fabrication method reduces casting defects such as shrinkage, porosity, and microstructural defects. The extrusion process is carried out by the application of graphite-based oil between samples and dies to reduce the friction and subsequent defects [65,78]. The lubrication allows the solid chips to be extruded by applying optimum extrusion pressure. For higher temperature extrusions, the glass powder is used as graphite-based oils are limited to low-temperature extrusion.

Various cutting processes such as lathe, sawing, milling, grinding, and drilling are used to obtain chips. The size of a chip is determined by the cutting method. Chip size can affect the final properties of the extruded products. Due to their small size, chips are easily oxidized [79]. On the other hand, oxidized chips are believed to increase the strength of extruded products [65,80]. Before the extrusion process, these chips are compacted to form billets. Compaction helps to reduce the entrapped in between chips during billet formation. Higher dislocation density renders the movement of particles for the plastic deformation, which will lead to delamination in between different layers. Preheating of billets is also performed to soften the billet structures. Higher ductility of billets assists in extrusion and plastic transformation. After all these pre-extrusion processes, billets are fully prepared for the extrusion. During extrusion, a large surface of billets is in contact with the die leading to higher friction. This friction is useful to break down the oxide layers due to strain generation. Extrusion temperature plays a vital role in the determination of the final properties of the product. The higher the extrusion temperature, the better the mechanical properties will be. High temperature leads to better tensile properties and diffusion [53].

Extruding complex geometries must be handled with extreme care because of the grain distortion affecting the final geometry [81–83]. In the hot extrusion process, the billets are reported to be transformed through compaction into extrudate. Either the direct or indirect hot extrusion method could be used for extrusion. Each method differs in properties and the area of application for the final products. Figure 3 explains the difference between both extrusion processes. Mostly, the direct hot extrusion method is used because of the higher compression in direct extrusion [84].

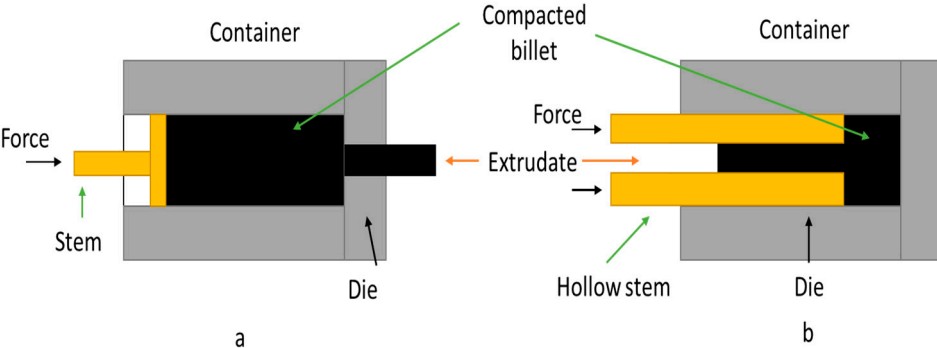

**Figure 3.** Difference between (**a**) direct and (**b**) indirect extrusion.

The large area of contact between the billets surface and the container wall is responsible for the higher friction force associated with direct hot extrusion [85]. Overman et al. [86] concluded that high extrusion temperature and low extrusion speed enabled the aluminum chip to be plastically deformed and flow at a high strain rate. The Box–Behnken experimental design was carried out by Krolo et al. [87], and it was reported that extrusion temperature was the most significant factor influencing the tensile strength.

### 2.3. Hot Extrusion Parameter

The extrusion time and temperature on hot extrusion process have effect on AA6061 aluminum alloy composite; as shown in Table 9, the average parameters change in different environments.

**Table 9.** Extrusion time and temperature for various Al alloy-based composites.

| Material<br>Parameters | AA6060<br>[88] | AA6061<br>[89] | AA6061<br>[90] | AA6060<br>[91] | AA6060<br>[92] | Average Parameter |
|---|---|---|---|---|---|---|
| Preheat temp (°C) | 500 | 550 | 550 | 450 | 500 | 450–550 |
| Preheat time (h) | 2 | 3 | 3 | - | - | 2–3 |

### 2.4. Parameter Optimization

The aim of optimizing a system is to make an optimal choice of the available resources for the process. This optimization process is a two-sided decision tool. Optimization considers the best use of the resources most efficiently and effectively to reach the highest achievable performance of the process under the given constraints and requirements. The concept is used to either maximize the desired factors or to minimize undesired ones. To maximize the desired factors means achieving the highest and maximum results or outcome without the cost constraints.

There are several methods to obtain optimal responses. For example, machine learning the particle swarm optimization (PSO) is another set of evolutionary algorithms that are commonly used for optimization problems. PSO algorithms are inspired by biological populations that exhibit both individual and social behaviors [93]. PSO works by enabling a group of particles (swarm) to traverse the search space in a semi-random manner. PSO algorithms identify the optimal solution through cooperation and information sharing among individual particles in a group. Moreover, Design of Experimental (DOE) and the Full Factorial method (Factorial) tools are easy to use and have found application in several experimental designs and investigations to optimize the response obtained [94,95].

This research does not take into consideration the interaction and optimization of different factors that may affect the properties of the material [96]. Therefore, the characteristics and mechanical properties of the material can be potentially explored using optimization tools [97]. By using machine learning models, one is assisted in the prediction of the outcome in processing materials, which will be helpful to minimize and limit the number of experiments, minimizing the time and energy consumption due to the high prediction rate.

### 3. Materials Properties

### 3.1. Mechanical and Physical Properties

The process of identifying specific properties of the material requires conducting and studying various characterization techniques. Physical, chemical, and mechanical properties are assessed through these characterization techniques. Table 10 presents the properties and characterization techniques to assess the properties.

**Table 10.** Material properties and type of testing.

| Properties | Testing |
|---|---|
| Mechanical | Tensile test, wear test, hardness test |
| Physical | Melting point, microstructure, thermal conductivity, density |
| Chemical | Oxidation resistance |

### 3.1.1. Density Test

The density test is used to evaluate the relationship between the densities of materials and chemical components. Density is known as the mass per unit volume with SI units of

kilograms per cubic meter (kg/m$^3$). Al-based MMCs are used to reduce the overall weight of automobiles, aircraft, marine ships, and spacecraft [2]. For the measurement of density, Archimedes principles are used. This method is an intuitive technique that can be used to find the density of materials. In this method, cleaned samples are weighed using weight balance. The sample is weighed under a vacuum for accurate measurements. The sample is then sunk into water or a different liquid. It must be ensured that no air is trapped in the sample. The weight of the root used to hold the specimen should be subtracted from the weight of the sample in water [98].

The density of the material can be calculated using formulas [99].

$$\text{Bulk density, } \rho_b = \frac{(W_d)}{(W_w - W_s)} \tag{1}$$

$$\text{Apparent porosity, } P_A = \left( \frac{(W_w - W_d)}{(W_w - W_s)} \right) \times 100\% \tag{2}$$

where
　　$W_d$ = weight of the dried sample.
　　$W_s$ = weight of the suspended sample in water.
　　$W_w$ = weight of the wetted sample in air.

Hot extrusion significantly altered the density, while the density of hot-rolled samples remained mostly unaffected [19]. Density increased with the increase in the reinforcement concentration [2]. Ravikumar et al. [47] reported that the density of the composite was increased with the increase in Al$_2$O$_3$ wt.% but decreased with the increase in wt.% of SiO$_2$ [66]. The incorporation of Al$_3$BC particles in the matrix materials contributed to the high dislocation density, load transfer, and grain refinement [100].

### 3.1.2. Ultimate Tensile Strength

Tensile properties of any material provide a deep insight regarding the mechanical properties of any material. The ultimate tensile strength is the load at which the specimen has reached the yield point and fractured. The addition of different reinforcements such as alumina, silica, copper oxide, silicon, zinc, and chromium particles in aluminum improves the yield strength and tensile strength while ductility is decreased [4,101,102]. Various studies reported an increase in tensile strength of the composites with an increase in the weight fraction of reinforcement [4,5]. With the addition of 1 wt.% nano SiO$_2$ particles, tensile strength increased by almost 24.8% [25].

Ferguson et al. [64] studied the addition of CuO nanoparticles in the aluminum matrix and reported a negligible effect on yield stress and failure strain. On the other hand, increase in CuO nanoparticle concentrations increased UTS (ultimate tensile strength). Nanoparticles have been successfully used in many different composites such as AA6061, AA6063, and AA7072, with the increase in mechanical properties [101].

### 3.1.3. Hardness Test

The purpose to conduct a hardness test is to measure the ability of the material to resist deformation and indentation. Such indentation can be made through abrasion, drilling, impact, scratching, and wear. Various techniques are used for the hardness test such as the Brinell, Knoop, Rockwell, or Vickers hardness technique [103]. Hardness is enhanced with the addition of ceramic reinforcements such as SiC, Al$_2$O$_3$, and B$_4$C to MMCs [104]. Ferguson et al. [64] reported strain-hardening exponent data for Al–0.5CuO, and Al–1.5CuO materials were higher than those of the respective alloys.

### 3.2. *Thermal Analysis*
### 3.2.1. Differential Scanning Calorimetry (DSC)

DSC is employed to measure and analyze the energy being transferred to and by the sample during the chemical and physical change. A very small quantity of sample is

required for DSC analysis. DSC can be performed on liquids and solids. DSC is useful in the determination of oxidative stability, thermal analysis, and degradation, along with moisture loss in the sample. During thermal analysis, endo- and exothermic peaks are obtained. Zhu et al. [105] evaluated Al–SiO$_2$ composite using DSC. When the heating rate was enhanced, reaction time was significantly lowered, and the exothermic peak shifted towards the higher temperature. The activation energy decreased with the increase in reinforcement concentration. Alam and Kumar [106] analyzed graphite nanoplatelet-based Al composite through DSC analysis. DSC showed the decomposition of samples around 600 °C. In addition, a complete decomposition was observed around 800 °C. Al matrix started to melt around 680 °C. An exothermic peak near 778 °C clearly showed the formation of Al$_4$C$_3$ in the composite samples. A more intense exothermic peak was obtained of Al-based graphite nanoplatelet composite in comparison with respective materials.

### 3.2.2. Thermal Gravimetric Analysis (TGA)

TGA is a tool to monitor the stability of the material and by monitoring the mass loss during heating of a sample at a constant heating rate. TGA analysis is used to measure the decomposition of any material. Mass change of the material is measured with respect to the temperature and time. TGA is used to know about the glass transition temperature and melting points in polymeric and metallic composites. Wang et al. [107] explained TGA analysis for graphene-modified copper nanoparticle Al-based nanocomposite. Wt.% of modified Cu nanoparticles was calculated using TGA. On the basis of these results, very little difference in both in situ and ex situ scenarios was observed. Seshappa and Anjaneya Prasad [108] used TGA to observe the mass loss of TiO$_2$/SiC-reinforced Al hybrid composite. It was observed that the formation of the oxide layer on the composite samples was delayed with the addition of reinforcements. With the increase in temperature during TGA, the diffusion rate of Al was enhanced.

### 4. Bibliometric Analysis

Bibliometric analysis is a set of techniques used for both qualitative and quantitative assessment of academic literature in any given subject topic. The bibliometric technique has been deployed by various studies to analyze the publication growth rate of any topic [109]. These techniques are found to be useful in analyzing the data to formulate a well-defined and sound research study under the objectives that will guide the researcher to contribute towards the high quality and meaningful research in the field of aluminum composites and recycling. The analyzed data are useful to assess the global status of research in nano-reinforcement materials.

### 4.1. Data

The data used in the present study were taken from Web of Science from 9 April to 3 May 2020. Web of Science is a comprehensive interdisciplinary and bibliographic database published by Clarivate Analytics. The search equation was TS = ("aluminum chips" OR "aluminium chips" OR "aluminum chip" OR "aluminium chip" OR "AA 6061" OR AA6061 OR "aluminium alloy 6061" OR "aluminum alloy 6061" OR "aluminium alloy" OR "aluminum alloy" OR "Waste aluminium" OR "aluminum waste" OR "aluminum industrial Waste" OR "aluminium industrial waste") AND TS = ("nano-composite" OR "Metal matrix composite" OR "hybrid nanocomposite" OR MMCs OR "hybrid composite" OR "copper oxide" OR CuO$_2$ OR CuO OR "copper dioxide" OR copper OR "silica oxide" OR SiO OR SiO$_2$ OR "silica dioxide" OR silica) AND TS = ("hot extrusion" OR "Solid-state recycling" OR "Solid-state process" OR "Solid-state recycling" OR consolidation OR "direct recycling" OR "powder processing")). The result revealed around 92 publications containing terms of aluminum recycling and reinforcement materials. This was used as a selection criterion for the analysis.

### 4.2. Country-Wise Analysis

Table 11 and Figure 4 below show the top 10 countries that published and engaged in the research related to nanocomposite materials. The frequency of citations represents the importance of this topic and the interest of researchers and industries for this particular topic. This kind of development was mainly related to the availability of funding from the Iran based agencies and the international collaboration between Iran and other countries. Iran published the highest number of research studies on aluminum-based composite recycling, followed by the USA, Spain, and Turkey.

**Table 11.** Publications related to nanocomposites.

| No | Country | Documents | Citations |
|----|---------|-----------|-----------|
| 1 | Iran | 14 | 373 |
| 2 | USA | 13 | 362 |
| 3 | Spain | 9 | 296 |
| 4 | Turkey | 9 | 220 |
| 5 | India | 8 | 71 |
| 6 | Japan | 8 | 321 |
| 7 | China | 8 | 121 |
| 8 | South Korea | 7 | 235 |
| 9 | Poland | 5 | 66 |
| 10 | Canada | 3 | 195 |

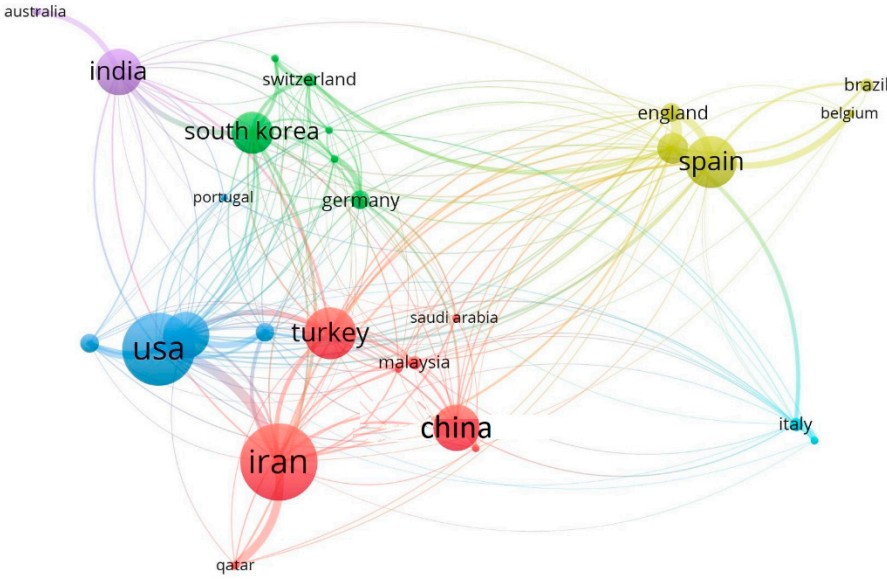

**Figure 4.** Publications related to nanocomposites.

### 4.3. Keyword Analysis

Table 12 and Figure 5 show the keyword that emerged from the analysis of this topic. The frequency of keywords represents the latest developments in a particular field. Keywords are behavior, mechanical properties, microstructure, hot extrusion, deformation, powder processing, strength, tensile properties, and copper. These keywords are relevant to recycling and reinforcements in this research area. As indicated by different colors in Figure 5, the keywords were categorized into six major clusters. Cluster 1 (red color) represents the characterization of mechanical properties, which involves studies on the mechanical behavior of elements with aluminum alloys as the largest node. This finding indicates that nano reinforcements in aluminum have been significantly used in different research studies.

**Table 12.** Keywords emerged from the analysis.

| No | Keyword | Occurrences |
|---|---|---|
| 1 | Behavior | 27 |
| 2 | Aluminum alloy | 26 |
| 3 | Mechanical properties | 26 |
| 4 | Microstructure | 23 |
| 5 | Extrusion | 17 |
| 6 | Mechanical properties | 16 |
| 7 | Hot extrusion | 15 |
| 8 | Metal matrix composite | 15 |
| 9 | Consolidation | 13 |
| 10 | Deformation | 13 |
| 11 | Metal-matrix composites | 13 |
| 12 | Powder processing | 13 |
| 13 | Strength | 12 |
| 14 | Tensile properties | 11 |
| 15 | Evolution | 10 |
| 16 | Metal matrix composites (MMCs) | 10 |
| 17 | Copper | 9 |

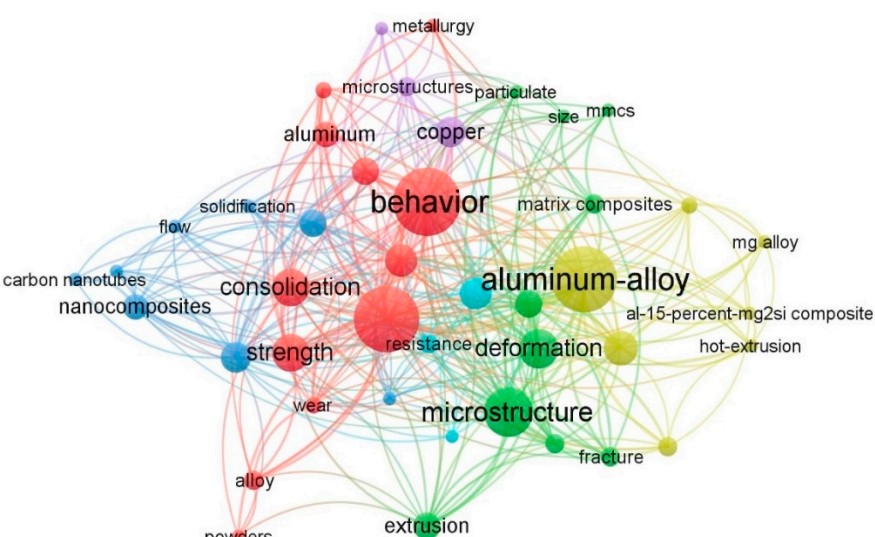

**Figure 5.** Keywords that emerged from the analysis.

Keywords and country-wise analysis have provided a fair idea about the scope of the addition of nanoparticles in the Al-based matrix materials. Various properties and fabrication methods were analyzed through this bibliometric analysis. The frequency of these keywords has indicated the scope of this review study. A number of research studies have focused on the recycling of Al, use of nano particles, and subsequent mechanical properties of these composites. Different research studies have analyzed the recycling and mechanical properties of Al alloys through hot extrusion. Furthermore, research studies have focused on the addition of copper oxide and silica nanoparticles to enhance the performance and mechanical properties of the Al-based composites.

## 5. Conclusions

This review has highlighted the significance of metallic matrix hybrid composite materials. Inclusion of silica and copper oxide nanoparticles significantly improve the strength and other mechanical properties. Nanoparticles as reinforcements are found to enhance the performance of the composites. Al-based matrix materials are widely used in the fabrication of metallic and hybrid composites. Al alloys are categorized in

various grades, and the choice of any Al-based alloy grade is dependent on the intended applications and required properties.

Al is extracted using conventional techniques and methods from ores. Conventional methods such as melting are used to recycle Al scrap. However, these methods are energy-intensive and waste the material due to the formation of oxide layer. Recycling of materials is required to address the environmental issues, pollution, and to save energy. Solid-state recycling is used to recycle Al without the melting process. The process has fewer steps with higher efficiency and consumes about 5% of energy used in the conventional extraction process. A few research studies are performed on the mechanical and physical properties of recycled AA 6061 Al alloy through solid-state recycling. This review covered various aspects involved in the solid-state recycling through hot extrusion. The factors influencing the final properties of extruded Al alloys have been discussed in detail.

Hybrid composites are formed by including two or more micro and nano reinforcements to the matrix materials. Various mechanical and physical properties of hybrid Al-based composites were analyzed with the inclusion of silica and copper oxide nanoparticles. Different research studies were discussed to see the effects of silica and copper oxide nanoparticles on Al hybrid composites. A detailed bibliometric analysis was carried out to reveal the publications regarding Al-based composites and nano reinforcements. This analysis was used as selection criteria for the review.

**Author Contributions:** Conceptualization, M.S.M.; writing—review and editing, M.H.R.; validation, S.M.T., C.N.A.J., A.H.M.A. and N.I.Z.; supervision, Z.L.; project administration, S.S. All authors have read and agreed to the published version of the manuscript.

**Funding:** This research was funded by University Putra Malaysia, Grant No. 9686400.

**Institutional Review Board Statement:** Not applicable.

**Informed Consent Statement:** Not applicable.

**Data Availability Statement:** Not applicable.

**Acknowledgments:** The first author thanks the School of Graduate Studies for financial support to study at Universiti Putra Malaysia. We acknowledge the use of facilities within the Centre for Graduate Studies, Universiti Tun Hussein Onn Malaysia (UTHM), and Sustainable Manufacturing and Recycling Technology, Advanced Manufacturing and Materials Center (SMART-AMMC), Universiti Tun Hussein Onn Malaysia (UTHM). We equally acknowledge the College of Engineering, Wasit University, Iraq, for research collaborations.

**Conflicts of Interest:** The authors declare no conflict of interest.

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
