# Peer review of "The Effects of CuO and SiO2 on Aluminum AA6061 Hybrid Nanocomposite as Reinforcements: A Concise Review"

_coatings, doi:10.3390/coatings11080972_

Round 1
Reviewer 1 Report
It is a quite nice review (which can be considered as a scholar one) that covers various aspects and mechanical properties of silica and copper oxide Al hybrid composite materials. The review has highlighted the significance of metallic matrix hybrid composite materials. It is shown that the inclusion of silica and copper oxide nanoparticles significantly improve the strength and other mechanical properties. Nanoparticles as reinforcements are found to enhance the performance of the composites. Authors also present the database search in addition to the straight literature review.
Please find below my more detailed questions and comments.
Title. I suggest to change (make more concrete) the title because not all Al-based ceramics are considered in the review but AA 6061 grade Al matrix mainly.
Table 11. It would be nice to give some references in the reference list to the documents listed in Table 11 (to link Web of Science search with the authors’ review and list of references). It will strengthen the content of the review.
Figure 4. I suggest to change “Peoples r china” just to “China” for simplicity. Otherwise one has to point out that other countries like Iran, for example, are also should be given with their full names (The Islamic Republic of Iran).
Sections 1.2. and 1.3 I did not quite get why so many place in the review is devoted to the Al-recycling? It is not the main point of the review.
Author Response
Dear Dr,
We appreciate the time and efforts of the reviewers in reviewing.
I attached the answer to your comments
best regards,

Reviewer 2 Report
The present study reports the article entitled “The Effects of CuO and SiO2 as Aluminum Ceramics-Based Reinforcements: A Concise Review”. It’s interesting work and to make this paper publishable, the author need to consider below comments:
-The figures are all ambiguous; hope in original version they’re more clear (high resolution). It’s not understandable anything in the present pdf file.
- The phrase “Aluminum Ceramics-Based Reinforcements” in title is unknown; Please change it with better words. Also, keywords are showing nothing (at least two first ones; e.g. aluminum alloys or bibliometric analysis); please replace with related words. It’s important for review papers. In the Abstract, hot extrusion should be mentioned somewhere as fabrication method.
-It’s a statistical research around the topic and can be interesting for readers; however, doesn’t have opinion piece of authors and possible future of this topic (ideas).
-example 1: the article does not include SEM, EDS, XRD, etc. micrographs or analysis results through the reviewed papers.
-example 2: the article does not talk about new technologies (e.g. additive manufacturing, 3D printing) for production of hybrid metal-ceramic composites. I’m sure there is many other methods.
Author Response

(The authors gave the same response as above.)

Round 2
Reviewer 2 Report
The article can be published in the present form.
This manuscript is a resubmission of an earlier submission. The following is a list of the peer review reports and author responses from that submission.
Round 1
Reviewer 1 Report
The article reviews the existing modern aluminum-based composites reinforced with copper oxide (CuO) and silicon oxide (SiO2), as well as the effect of copper oxide and silicon oxide additives on such operational and mechanical properties as strength, hardness, viscosity, and density. Separate chapters of the article are devoted to the influence of nanoscale ceramic powders on aluminum alloy and the technology of pressing the alloy by hot extrusion.
The relevance of the work is confirmed by the wide application of the developed aluminum-based composites in various fields of science and technology, including the aerospace, marine and automotive industries. The paper presents unique data of aluminum-matrix composite materials with different content of CuO and SiO2 reinforcing ceramics (3 wt.%, 10 wt.%, 30 wt. % ), which open up new horizons in the use of composites.
Of particular scientific value are the results of studies of composite materials with additives of reinforcing nanoscale powders CuO and SiO2. With minor additives (3 wt. % ) it turned out to significantly increase the value of the composite hardness due to the uniform distribution of ceramic silica nanoparticles. Even when adding 1 weight. The mechanical and physical properties of the composite were significantly improved, and the hardness and tensile strength of the nanocomposite were increased by 41.8% and 24.8%.
The paper presents data on the production of composite materials based on aluminum by hot extrusion and presents data on the optimization and modeling of the process, in which the temperature and heating time of the workpiece varied.
Comments:
1) Metallographic studies of hybrid aluminum-matrix composites reinforced with CuO and SiO2 ceramics are not presented in this paper. Microscopy images taken from the surface of the material sections would allow us to quantify the size of the reinforcing filler conglomerates, the uniformity of distribution in the aluminum matrix, and the porosity estimation by metallographic method.
2) The paper does not address the methods of X-ray phase studies of hybrid aluminum matrix composites reinforced with CuO and SiO2 ceramics. Such studies provide important information about the stability of the production technology and the quality of the initial components and the final composite.
3) It was worth considering the inclusion in the section of the obtained properties of the relationship between the change in the content of the filler of reinforcing ceramics CuO and SiO2 in the composite and the coefficient of thermal linear expansion of the composites, which will form a complete table of material characteristics (including performance).
The article is developed and executed qualitatively, has a high scientific potential and a valuable bibliometric analysis of publications on the topic of aluminum-matrix hybrid composites. Given the large amount of research done by the author, the comments made do not reduce the value of the work and do not have a significant impact on the overall despite the comments provided, it is recommended for publication.
Author Response
thank you for your patience

Reviewer 2 Report
The title of the manuscript is “The effects of CuO and SiO2 as aluminum ceramics-based reinforcements: A Concise Review”. And the authors claimed that the aim of the present review manuscript is “The present work focuses on the development of nano silica oxide and nano copper oxide particles reinforced Al based composite (introduction)”. Considering the aim of the manuscript the authors proposed, contents (quality) and structures should be improved remarkably.
- First of all, the structure of the manuscript is somewhat confusing and the content is not sufficient as a review article. In addition, I found several wrong information in the manuscript.
(1) Much work has been done on the fabrication of AMCs (aluminum matrix composites) with the addition of metal oxide such as SiO2, CuO, TiO2, ZnO etc into aluminum and/or its alloys in the solid and/or liquid state. Basically, the in-situ reaction (solid state and/or liquid state) occurs between MO (metal oxide) and aluminum to form alumina (Al2O3…) and the metal reduced to aluminum matrix. Thus, the control of reaction is key parameter to make sound composites, since the size, vol% and type of resultant alumina and the degree of reaction strongly dependent on the reaction. However, it is not included in the manuscript at all.
(2) In section 1.4.1.1 and 1.4.1.2, I got the impression that the authors confused the addition of SiO2 with Al-Si alloy. Similarly, it is the same for CuO and Al-Cu alloy. Due to this reason, there are several wrong information in these sections (including table 6 and 7).
(3) The contents of Tables 1, 2,3,4,5,8,10 are text-book level for general aluminum alloys and MMCs, not directly related with CuO and/or SiO2 reinforced AMCs.
Author Response
thank you for your patience

Reviewer 3 Report
The reviewed paper entitled “The Effects of CuO And SiO2 as Aluminum Ceramics-Based Reinforcements: A Concise Review” concerns the review of the state of knowledge in the area of hybrid composites consists of Al (matrix) and nanoparticles CuO and SiO2 (reinforcement).
The work, written in a very careful manner, on the basis of extensive and up-to-date literature, presents the entire "technological cycle" from aluminum recycling, through the techniques of producing composites (taking into account various techniques and process parameters) to testing the properties of the obtained materials.
The work is valuable and can be useful in the design of hybrid composites, especially in relation to the discussed compositions.
Author Response
thank you for your patience

Reviewer 4 Report
The title of the review highlights the effects of CuO and SiO2 on reinforcements of aluminium alloys. The authors give a good introduction to aluminium alloys in general, its production and recycling possibilities. However, the core topic that is announced via the title is only adressed on 4 of 17 pages. Therefore, I suggest to give more details on the CuO, SiO2 reinforcement topic. The other addressed topics are relatively general and are only slightly above book knowledge. Also much of the given information was taken from other reviews on aluminium alloys. It seems therefore questionable how much additional information is given. It would be helpful to reduce the amount of pages to only including the core topic.
Please find below my additional comments.
Table 3: wt.% ?
Table 5 can be given as listing within the text. See also 'K2o' should be 'K_2O'.
l 217-218: 'The addition of silica .. ' This sentence seems to be erroneous. Do you mean '(..) during casting of the alloy.'? From a thermodynamic point of view, silica is chemically not stable against liquid aluminium. I suggest to add a related discussion here or before line 236.
l 295 'while' --> 'While'
l 316, Figure 3, Mark a) and b) in the figure caption.
l 362 'g/m³' --> 'kg/m³'
l 394 'UTS': term not explained
l 411 'activation energy': Please explain in more detail 1) activation energy for which process and 2) the method of its determination.
l 423 'Thermal Galvanometric Analysis (TGA)' I guess you mean 'Thermal gravimetric analysis' normally combined as DSC/TGA or DTA/TGA.
Reference 31: Please give complete bibliographic information, e.g. university, publisher etc.
ref [56]: 'no xxxx' Delete or add the actual number.
Author Response
thank you for your patience

Reviewer 5 Report
The authors analyzed the strengthening effect of nano-SiO2 and CuO reinforced aluminum matrix composites in some literature. The solid state recycling and fabrication methods of aluminum are presented. Here are some comments for further improvement of the paper.
1. What does the article mainly describe? The strengthening effect of nanophase? The method of preparation and testing? The problems of aluminum recycling? The focus is not clear.
2. What are the obvious conclusions about the analysis of the references? What is the significance of this part?
3. What is the relationship between the discussion of hot extrusion and the CuO and SiO2 reinforced phases?
4. What is the relationship between SiO2 and CuO and aluminum recycling as mentioned in the text?
5. Silica oxide and silica, two statements to retain one.
Full text: wt% and wt.% keep one
Table 7, Table 9, Table 10, Figure 1, Figure 2, remove shows.
P1, line 29, SiO2.
P2, Table 1, Matrix material, ceramic some are plural and some are not.
P2, line 60, copper oxide-based?
P2, line 62, these particles are.
P2, line 65, lead to a decrease.
P3, Table 2, wrong density units.
P3, Table 3, is it wt.% or vol.%?
P6, Table 5, K2O.
P7, line 225 and Table 6, α-Al2O3.
P7, lines 227, 234, 235, silica-based Al composite?is Al-based.
P8, Table 6, What is the difference between Al-Si and Al-SiO2?
P8, line 236, Copper (CuO) does not correspond.
P9, Table 7, not Cu-based but Al-based.
P12, Figure 3, explain which is direct extrusion and which is indirect extrusion.
P12, Table 9, units of temperature; format of AA6060, etc.
P13, line 359, remove This.
P14, line 385, remove is.
P14, line 399, can be created?
P14, line 413, Zhu et al.
P16, Figure 1 changed to Figure 4.
P16, line 477, Figure 7 to Figure 5.
P17, Figure 2 to Figure 5; remove shows.
Regarding subheadings.
P2, 2.1 changed to 1.1.
P6, two 1.4.1.
P11, no 2.2 why write 2.1.
P14, no 3.2 jumped directly to 3.3.
P15, how can there be 4.1.1 without 4.1; why write 4.1 without 4.2?
Author Response
thank you for your patience

Round 2
Reviewer 2 Report
Compared to the original manuscript, it cannot be found improvement in the revised manuscript.
I strongly recommend to the authors to delete sections 1.4.2.1 and 1.4.2.2. The authors are still confusing (i) matrix alloy (Al alloy, Al-Si alloy, Al-Cu alloy and Cu alloy) with reinforcement (SiO2, CuO, ….), and (ii) in-situ reaction by solid-state with liquid-state. Due to this reason, there are several wrong information along with unnecessary informantion in these sections (including table 6 and 7).
The contents of Tables 1, 2,3,4,5,8,10 are text-book level for general aluminum alloys and MMCs, not directly related with CuO and/or SiO2 reinforced AMCs.
Author Response
Thank you very much we appreciate it

Reviewer 4 Report
l 217: The sentence was not corrected, it is unclear what fluidity you mean. I guess the sentence should be 'The addition of silica particles in the Al matrix is believed to be responsible for the fluidity of the liquid alloy during casting.' Please note that the term 'matrix' is normally not used for a liquid matrix.
section 3.2.1, l 411 'activation energy'. You can not simply say DSC is used to measure activation energy. This statement is to general. Also it must be explained in more detail. If you calculate the integral of a DSC peak, the energy would refer to an enthalpy of reaction not an activation energy. An activation energy is always linked to a certain chemical reaction or process. It is unclear what is meant here.
line 423, 'Thermalgalvanometric Analysis': This analysis does not exist. You mean most probably 'Thermal Gravimetric Analysis'. Please correct.
reference 31: The university name is written on the first page of Valentine Richter-Trummer's dissertation; it is 'Faculdade de Engenharia da Universidade do Porto'.
Author Response
Thank you very much we appreciate it.

Reviewer 5 Report
The paper can be published now.Author Response
Thank you very much we appreciate it.
Round 3
Reviewer 2 Report
First of all, the title of the manuscript is “The effects of CuO and SiO2 as aluminum ceramics-based reinforcements: A Concise Review”. And the authors claimed that the aim of the present review manuscript is “The review covers various aspects and mechanical properties of silica and copper oxide Al hybrid composite materials. (introduction)”. Considering the aim of the manuscript the authors proposed, I read the manuscript several times and I point out the misleading statements, incorrect information, confusing statement etc; but it is not all.
- section 1
(1) line 61 : “ Due to the excellent mechanical and physical properties of CuO and SiO2, these particles are ~ “ ---> What is the excellent mechanical and physical properties? Please provide them clearly as a table.
(2) line 66-67 : “ Cu addition reduces the melting point and results in the formation of Al2Cu phase which enhances the strength of Al matrix” ---> This is not related with the CuO. This is general phenomena for Al-Cu alloy.
(3) Table 2 : it contains many wrong information. Please check it.
(4) fig.1 : almost unnecessary. What do the authors want to say by this figure? Who categorized this figure? By yourself? Otherwise put the reference.
(5) Table 4 : unnecessary. it contains many wrong information. Please check it.
(6) line 123-124 : “ The main issue in the recycling of Al is to maintain the chemical composition as alloys of Al contain various amounts of alloying elements and impurities” ---> The main issue in the recycling of Al is removal of inclusion such as oxide particle/bifilm, Fe removal and control of pore. Alloying elements can be controlled after refining process.
(7) line 139 : “ Moreover, a considerable amount of material is lost due to oxidation during the melting process” --- > What does it mean ?
(8) Table 5. Unnecessary and no meaning. All of the nano particles such as ceramics and intermetallics can be used as reinforcements for Al matrix composites (AMCs). Please remove it, otherwise provide physical and mechanical properties of each reinforcement along with reference.
- section 1.4.2.1
(1) line 220 : “Silica combines with aluminum to make the alloys 220 heat treatable [53].” --- > I cannot find that kind of statement in ref [53]. Please check it. It is the same as table 6.
(2) line 221 : “ This material has various applications in the pharmaceutical industry, printing toners, electronic chips/parts and in the treatment of cancer.” Please remove the statement. This is just for SiO2 not related with SiO2 particle reinforced AMC.
(3) line 225 : ref [65] does not contain the properties of Al/SiO2.
(4) Table 6 : ref [55] contains the liquid state processing for Al/siO2 composites. Not related with this topic. Please remove it.
- section 1.4.2.2
(1) line 236-244. This paragraph explains the Cu alloy matrix composites, NOT related with Al/CuO. Please remove it. In addition, ref [56] does not related with even Cu alloy.
(2) line 246-256. This paragraph explains the Al-Cu alloy matrix composites, NOT related with Al/CuO. Please remove it. It is the same for table 7.
(3) Among the references in this section, there is only one references for Al/CuO composites; ref [66]. Please think about the reason why there are very few Al/CuO composites in the literature. Actually, CuO is very dangerous when added into Al due to fast reaction with enormous heat release. CuO is normally used as a dopant to control reaction when AMCs are fabricated by in-situ process such as SPS process.
- section 1.4.2.3
MMCs including AMCs have been studied for more than three decades. The purpose of MMCs is tailoring of the properties (not maximize). Up to now much work has been done and reported for fabrication of sound composites. In addition, prediction of materials properties by various mathematical model has been developed such as micro-mechanics approach and shear-lag model. In this regard, table 8 is not necessary at all.
- section 3
(1) In this section, the authors describe the test method, not test results related with SiO2 and/or CuO reinforced AMCs. Test method is general for all kind of materials not directly related with AMCs.
(2) regarding density, what do you think is the purpose to measure density for MMCs?
(3) line 384-386 : “The addition of alumina, silicon, zinc and chromium particles in aluminum improves the yield strength and tensile strength while ductility is decreased.” --- > What does it mean for Si, Zn, Cr ~~~?
(4) line 384-386 : “To avoid the reaction between the reinforcements and matrix material, reinforcements must be heated” --- > I cannot understand and even I cannot agree to the statement. What does it mean?
(5) line 389-390 : “With the addition of 1 wt% nano SiO2 particles, tensile strength increased by al-389 most 24.8% [25]” --- > However, you said opposites in line 65-66.
(7) line 416-417 : “An exothermic peak near 778° clearly showed the formation of Al4C3 in the composite samples.” --- > This result is not related with SiO2 and/or CuO reinforced AMCs. This is very important for carbide particle reinforced AMCs. Please remove ref [108].
(8) As you categorized DSC and TGA for “thermal properties”. It is not “thermal properties”, it is “thermal analysis”. Normally, “thermal properties” means “CTE” and “thermal conductivity” for MMCs. “thermal analysis” is performed for detect the reaction between matrix and reinforcements.
(9) Please add “thermal properties”, “wear properties” and “elastic properties” of SiO2 and/or CuO reinforced AMCs. The aforementioned properties are most important for AMCs. In general, the tensile properties of aluminum alloys are tailored by alloying elements such as Cu, Mg, Mn, Zn etc. However, the aforementioned properties are not improved by addition of alloying elements. So, the purpose of the MMCs is generally to improve the aforementioned properties not improving tensile properties.
Author Response
We appreciate the time taken by reviewer for such detailed reviews. The valuable insights and suggestions helped us to improve the manuscript further
Best regards,
